# Combined metabolomics and proteomics analysis of vascular cognitive impairment in hypertensive rats induced by endothelial injury

Xiao Zhang[1], Surui Chang[1,2], Jiangang Liu[1]*, Dongling Wang[1]

1 Xiyuan Hospital, China Academy of Chinese Medical Sciences, National Center for Clinical Cardiovascular Disease of Traditional Chinese Medicine, Beijing, China, 2 Haikou Traditional Chinese Medicine Hospital, Haikou, Hainan, China

* liujiangang2002@sina.com

## Abstract

### Aim

Based on metabolomics and proteomics research, we investigate the molecular biological mechanisms underlying vascular cognitive impairment (VCI) in rats induced by hypertension combined with endothelial damage, aiming to identify proteins or metabolites associated with key metabolic pathways.

### Methods

SPF hypertensive rats (SHR) were used to damage the common carotid artery endothelium with microcurrent to induce vascular cognitive impairment model in rats (Model group), SHR rats (SHR group) and SPF normotensive Wistar-Kyoto rats (WKY) with the same genetic background were used as sham operation groups (no current stimulation after dissection), with 10 rats in each group. Morris water maze, PNT experiment and SPT experiment were used to detect the learning and memory ability of rats. TMT quantitative proteomics technology combined with liquid chromatography tandem mass spectrometry (LC-MS/MS) was used to detect the difference in metabolites and proteins in brain tissue of rats with vascular cognitive impairment.

### Results

From Day 1–5, compared with the WKY group, both the Model and SHR groups exhibited shortened incubation periods and slower average swimming speeds ($P<0.01$); On day 6, compared with the WKY group, the Model group showed a significant decrease in the number of platform crossings ($P<0.05$), time spent in the target quadrant ($P<0.05$), and total distance traveled in the target quadrant ($P<0.05$). Pathological results revealed that, compared with the WKY group, the SHR group and Model group exhibited a decrease in the number of glial cells and

**Data availability statement:** All relevant data are within the manuscript and its Supporting Information files.

**Funding:** This study was supported by Major Increase and Decrease Projects at Central Level (No.2060302). The funders had no role in study design, data collection and analysis, decision to publish, or preparation of the manuscript.

**Competing interests:** The authors have declared that no competing interests exist.

neurons in the hippocampal tissue, with relatively loose arrangement of cell bodies and lighter Nissl staining. In the Model group, some hippocampal neurons changed from a granular to a powdery appearance, with nuclear pyknosis, accompanied by cellular edema and necrosis. The metabolomics results showed that there were 437 significantly different metabolites between the Model group and the WKY group, 128 (+) and 309 (-); there were 449 significantly different metabolites between the Model group and the SHR group, 119 (+), 330(-). The differential metabolites in each group were mainly concentrated in metabolic pathways such as alanine, aspartate and glutamate metabolism, arginine and proline metabolism, and amino acid biosynthesis. The proteomics results showed that compared with the WKY group, there were 141 differentially expressed proteins in the Model group, 55 (+), and 86 (-). Pathways with higher connectivity in the action network include "glycine, serine and threonine metabolism", "phenylalanine metabolism", etc.; compared with the SHR group, there were 28 differentially expressed proteins in the Model group and 15 (+). 13 species (-). The significantly enriched pathways are "focal adhesion" and "relaxin signaling pathway".

## Conclusion

Rats with hypertension combined with endothelial injury have behavioral disorders, which are similar to the causes and signs of clinical vascular cognitive impairment; the biological mechanisms manifested in the model involve amino acid metabolism disorders, energy metabolism, relaxin signaling pathway and impaired focal adhesion function. Characteristic metabolites in the model group, such as ATP, cAMP, Creatine, Cyanocobalamin, Dopamine, Serotonin, Tryptophan, Uric acid and Vitamin B1 decreased, while GABA, Glucose, Isocitrate and Malate increased.

## Background

Vascular cognitive impairment (VCI) is caused by the dysfunction of the neurovascular unit and cerebral blood flow regulation disorders, involving damage to the endothelial cells of the vascular wall and the glial cells outside the vascular wall, oxidative stress and immune inflammatory response, which ultimately leads to neuronal damage, which is the key factor in the dysfunction of the neurovascular unit [1]. Clinical manifestations include communication disorders, memory loss, distraction, personality changes, etc. Vascular dementia (VD) is its severe stage, which refers to dementia caused by a series of cerebrovascular factors (ischemia, hemorrhage or acute and chronic hypoxic cerebrovascular disease, etc.) that cause brain tissue damage. Epidemiology shows that the number of dementia patients doubles approximately every five years, VD accounts for at least 20% to 40% of all dementia diagnoses, and the prevalence of VD in individuals aged 50 years and above is about 116/10,000 people [2,3]. It is currently believed that one of the most important types of pathophysiological factors leading to vascular dementia is ischemic and hypoxic hypoperfusion [4].

Due to hypertension and arteriosclerosis, the important parts of the cerebral cortex involved in cognitive function and the brain tissue that is more sensitive to ischemia and hypoxia are in a state of ischemic hypoperfusion for a long time, which causes delayed necrosis of neurons in these parts and gradually leads to cognitive dysfunction. At present, most of the vascular cognitive impairment models are made by vascular occlusion and vascular embolization [5]. Although the conditions of brain hypoperfusion can lead to dementia-like cognitive dysfunction, they are still quite different from the causes of vascular dementia in clinical patients. Hypertension is a common factor in VCI [6]. When cerebral blood vessels are exposed to high pulsatile pressure and flow, microvascular damage occurs, resulting in white matter damage and cortical detachment, which leads to the occurrence of VCI and VD [7].

We used hypertensive rats combined with microcurrent to injure their carotid endothelium to cause thrombosis, resulting in repeated cerebral ischemia and hypoxia-induced vascular cognitive impairment models. Metabolomics and proteomics are used to focus on the changes in small molecule metabolites and protein composition in the body, hoping to fully reflect the metabolic network of VCI rats and reveal the complex regulatory network of protein levels during the pathogenesis of VCI. Through joint analysis, the interaction and regulatory relationship between proteins and metabolites can be found. Therefore, in this study, we explored the formation mechanism of hypertension combined with carotid endothelial injury to vascular cognitive impairment model rats based on metabolomic and proteomic technologies, proposed a molecular biology change mechanism model, and jointly analyzed the expression data of differentially expressed proteins and metabolites to find out the differentially expressed proteins and metabolites that have synchronous rule of change, and then performed personalized analysis by combining with correlation coefficient matrix heatmap, correlation analysis clustering heatmap and correlation coefficient regulatory network diagram for personalized analysis. This provides a data basis for subsequent in-depth experiments and analysis, and explores possible intervention targets or signal pathways for VCI and VD.

## 1. Materials and methods

### 1.1. Experimental animals

The animal experiment for this study received Institutional and Animal Ethics approval from Ethics Committee of Xiyuan Hospital, China Academy of Traditional Chinese Medicine(№: 2021XLC002–2). Spontaneously hypertensive rats (SHR) and Kyoto Wistar rats (Wistar-Kyoto, WKY) were male, 13–14 weeks old, 180-200g, and purchased form Beijing Viton Lihua Laboratory Animal Technology Co. and kept in the standard laboratory conditions in a temperature-controlled environment (22–25°C) with 12-hour light/dark cycles. The rats were acclimatise for a week prior to the study and were provided ad libitum access standard normal chow from Beijing Keao Xieli Feed Co..

### 1.2. Experimental instruments

QTRAP5500 mass spectrometer, produced by Shanghai Aibocaisi Analytical Instrument Trading Co., Ltd.; Nexera X2 LC-30AD high performance liquid chromatograph, produced by Shimadzu (Shanghai) Laboratory Equipment Co., Ltd.; Waters Acquity UPLC HSS T3 column, produced by Waters Technology (Shanghai) Co., Ltd.

QE HF-X high-resolution orbitrap liquid-mass spectrometer, provided by Thermo Fisher Scientific (China) Co., Ltd.; EASY-nLC 1200 chromatograph, provided by Thermo Fisher Scientific (China) Co., Ltd.; 89870 reversed-phase HPLC column, provided by Thermo Fisher Scientific (China) Co., Ltd.; SC601 EASY-Column™ capillary HPLC column, provided by Thermo Fisher Scientific (China) Co., Ltd.; MultiSkan FC MultiskanSkyHigh full-wavelength microplate reader, provided by Thermo Fisher Scientific (China) Co., Ltd.; Pierce™ High pH Reversed-Phase Peptide Fractionation Kit, provided by Thermo Fisher Scientific (China) Co., Ltd.; TMTpro™ 16plex labeling reagent (TMTpro plex kit), provided by Thermo Fisher Scientific (China) Co., Ltd.

## 1.3. Model establishment and grouping

The model rats of vascular dementia were prepared by applying microcurrent stimulation to the bilateral common carotid arteries. Prior to the surgery, rats were fasted for 12 h followed by routine aseptic technique. Anesthesia was induced via intraperitoneal injection of 2% pentobarbital sodium. After anesthesia, the rats were fixed in supine position, with neck skin depilated and disinfected with iodine tincture. A longitudinal midline incision (1.5–2.0 cm) was made to expose the bilateral common carotid arteries. Blunt dissection was performed to separate the neck skin and muscle layers, followed by bluntly dissecting the common carotid arteries and their accompanying nerves cranially and caudally. Two sections of surgical line were clamped and passed through the bottom of the common carotid artery and then arranged in parallel. The two ends of the surgical threads were placed on the left and right sides of the blood vessels respectively, and the surgical threads was lifted to place the gasket under the blood vessel. Turn on the power, turn the timing and stimulation switches to the ready position, and prepare the stimulation electrodes and temperature probe. The power plug was connected to a 220V power supply, and the digital display was reset before each operation. The zero adjustment knob was used to set the voltage to 80mV, and the current timing switch was adjusted to deliver 0.3A microcurrent stimulation for 3 minutes. Then place the electrodes on the bilateral common carotid arteries of SHR rats, and shock the common carotid arteries of rats at the set power and time until the instrument automatically alarms to indicate thrombosis.

In the WKY group, blunt dissection of bilateral common carotid arteries was performed with suture placement only (no electrical stimulation). After the procedure, the silicone pad and surgical sutures were removed from beneath the vessels, followed by layered closure of the incision. All the animal were kept warm through out the surgery to avoid hypothermia. For post surgery care, close monitoring were done and the rats were given penicillin (20 IU/kg) via infection for 3 days to minimize the risk of infection. Morris water maze experiment was performed 1 week after modeling, and rats with behavioral cognitive impairment were screened as the Model group by comparison with the WKY group. Rats with cognitive impairment were identified by comparison with WKY controls and designated as the Model group (SHR + vascular injury composite model). Additionally, SHR rats (SHR group) and Wistar-Kyoto rats (WKY group) with identical genetic background served as sham-operation controls, with 10 rats in each group.

## 1.4. Rat behavioral test

The Morris water maze (MWM) was used to evaluate the spatial learning and memory function of the model rats [8]. The equipment includes a circular pool (50 cm high and 120 cm in diameter), an automatic camera, and a computer video analysis system. The pool is divided into four quadrants in a clockwise direction, and a platform is placed in the center of the first quadrant. Clean water is poured about 1 cm above the platform, and ink is added to the water to hide the platform. The water temperature is maintained at $(21 \pm 2)°C$. The indoor environment and light source are kept quiet during the experiment. Rats were acclimated in the MWM laboratory one day before the experiment.. During the experiment, rats were placed into the water head-down facing away from the pool wall, with each trial lasting 60 seconds and a rest interval of at least 30 minutes between trials to ensure adequate recovery. The experiment lasted 6 days total. The first 5 days were the Place navigation test (PNT) to evaluate the spatial learning ability of the rats, while the 6th day was Spatial Probe Test (SPT) to evaluate memory retention.

## 1.5. Pathological structural observation of synapses in the cerebral cortex and hippocampus

### 1.5.1. Hematoxylin and Eosin (H&E) staining. 
The rats were sacrificed by intravenous injection of an excessive amount of sodium pentobarbital, and then the brain tissues were excised. Immerse the rat brain tissue in 10% neutral formaldehyde solution for fixation, followed by embedding, deparaffinization, and gradual ethanol dehydration. Stain with 1% hematoxylin and eosin, and mount with neutral gum for observation under an optical microscope. Use an HE staining kit to stain and capture images at ×200 and ×400 magnifications in the hippocampal CA1 and CA3 regions, selecting neurons with uniform staining and relatively intact structure to showcase the pathological morphology of the brain tissue.

**1.5.2. Nissl staining.** Neuronal Nissl bodies in hippocampal CA1 and CA3 regions were observed under light microscopy at ×200 and ×400 magnification to examine neuronal structural integrity and assess neuronal damage. The toluidine blue method was performed as follows: Firstly, fresh tissues were fixed in 10% formaldehyde solution followed by routine dehydration and paraffin embedding. Secondly, paraffin sections (5 μm thickness) were dewaxed and hydrated to water. Thirdly, sections were incubated with Nissl staining solution at 50–60°C for 20–40 minutes. Fourthly, after brief washing with distilled water, sections were rapidly differentiated in 95% ethanol, dehydrated in absolute ethanol, cleared in xylene, and mounted with neutral resin.

## 1.6. Metabolomics detection

Chromatographic conditions: During the entire analysis process, the sample was placed in an autosampler at 4°C, and the sample was separated using a Shimadzu Nexera X2 LC-30AD ultra-high performance liquid chromatography system (UHPLC) using a T3 chromatographic column. Mass spectrometry conditions: Each sample was detected using electrospray ionization (ESI) in positive ion (+) and negative ion (-) modes. The samples were separated by UPLC and analyzed by mass spectrometry on a 5500 QTRAP mass spectrometer, using a HESI source for ionization. Use MRM mode to detect the ion transition to be measured.

## 1.7. Protein testing

TMT proteomics methods were used to detect and analyze proteins in brain tissue. For each sample, appropriate amounts of peptide fragments were taken for chromatographic separation using the nanoflow rate EasyLC1200 chromatography system. After peptide separation, DDA (data-dependent acquisition) mass spectrometry was performed using a Q-Exactive HF-X mass spectrometer. The analysis time is 60 minutes, detection mode: positive ion, precursor ion scanning range: 300-1800m/z, primary mass spectrometry resolution: 60000@m/z200, AGC target: 3e6, primary Maximum IT: 50ms. Secondary mass spectrometry analysis of peptides was collected according to the following method: triggering the acquisition of 20 highest intensity precursor ion secondary mass spectra (MS2scan) after each full scan, secondary mass spectrometry resolution: 15000@m/z200, AGC target: 1e5, Level 2 Maximum IT: 50ms, MS2ActivationType: HCD, Isolation window: 1.6m/z, Normalized collision energy: 32. Proteome and Discoverer software were used for qualitative and quantitative calculations of TMT-labeled proteomic data.

## 1.8. Statistical analysis and processing

The obtained data were expressed as Mean±SD. The data conforming to normal distribution and homogeneity of variance were analyzed by One-way ANOVA. Kruskal-Wallis test was selected when normal distribution was not met. Bonferroni test was used for multiple comparisons between groups, while Dunnett's T3 test was applied when variance homogeneity was violated. Orthogonal Projections to Latent Structures Discriminant Analysis (OPLS-DA) is a statistical method of discriminant analysis. This method uses partial least squares regression to establish a relationship model between the expression levels of metabolites and the sample categories, so as to achieve the prediction of sample categories. This method is revised on the basis of Partial Least Squares Discriminant Analysis (PLS-DA), filtering out the noise irrelevant to the classification information, and improving the analytical ability and effectiveness of the model. Calculate the Variable Importance for the Projection (VIP) for OPLS-DA, using $VIP > 1$ as the screening criterion to initially identify differential metabolites between groups. Select metabolites with $VIP > 1$ and $P < 0.05$ as those with statistically significant differences. Compute the fold change (FC), and screen for significantly differential proteins using $P < 0.05$ and $FC > 1.2$ or $FC < 1/1.2$. Perform functional analysis on differential metabolites or proteins and construct relevant enrichment pathways. The data were plotted using Origin software, and bioinformatics data were analyzed with Perseus and R software. Behavioral data were analyzed using SPSS 26.0 software, with statistical significance defined as $P < 0.05$.

## 2. Results

### 2.1. OPLS-DA

OPLS-DA can achieve dimensionality reduction by extracting latent variables directly related to sample categories, while separating and removing orthogonal noise to reduce interference, thereby more clearly revealing inter-group metabolic differences and enhancing the interpretability and predictive stability of the model for sample classification. $R^2X$ and $R^2Y$ represent the interpretation rates of the model for matrix X and matrix Y respectively; $Q^2$ represents the prediction ability of the model. The closer the values of $R^2$ and $Q^2$ are to 1, the more stable and reliable the model is. When $Q^2 > 0.5$, it indicates that the model has a good prediction ability, while when $Q^2$ is less than 0.5, it shows that the model's prediction ability is poor.

### 2.2. Behavioral comparison between rats in each group

The MWM test is the most classic experiment for detecting the cognitive function of VCI rats. Through the behavior of rats searching for a hidden platform in water, it reflects their spatial localization, learning and memory abilities. The results showed that from D1 to D5, compared with the WKY group, the Model group and SHR group exhibited significantly shorter latency ($P < 0.05$, $P < 0.01$) and faster average swimming speed ($P < 0.01$). On D6, the platform was removed to initiate the spatial exploration test. Rats were placed into the pool from the opposite side of the original platform quadrant, and the time spent searching in the target quadrant and the number of entries into this quadrant were recorded as indicators of spatial memory. Compared with the WKY group, the Model group demonstrated significantly fewer platform crossings ($P < 0.01$), reduced target quadrant search time ($P > 0.05$), and increased total distance traveled in the target quadrant ($P > 0.05$) (Fig 1).

### 2.3. Pathological staining

**2.3.1. Pathological changes in the hippocampus of rats by HE staining.** The results indicated that in the WKY group, the hippocampal tissue structure was normal with tightly and orderly arranged neuronal cells showing uniform staining, regular plump morphology, and normal nuclear staining. In contrast to the WKY group, both the SHR and Model groups exhibited relative reduction in neuronal cell quantity within the hippocampal tissue structure, accompanied by glial cell nuclear deformation and pyknosis, as well as relatively loose arrangement of glial cells (Fig 2).

**2.3.2. Pathological changes of Nissl staining in the hippocampus of rats.** The results showed that in the WKY group, neurons in the hippocampal CA1 and CA3 regions were arranged regularly. Nissl bodies and cell nuclei appeared blue-purple, with Nissl bodies displaying granular morphology and abundant quantity. In contrast, both SHR and Model groups exhibited relatively loose neuronal arrangement in the hippocampus, with lighter Nissl body staining. In the Model group, some hippocampal neurons transformed from granular to powdery morphology, accompanied by nuclear pyknosis, cellular edema, and necrosis (Fig 3).

### 2.4. Comparison of differential metabolites in rat brain tissue

The aforementioned detections revealed structural damage of hippocampal neurons and impairment of spatial memory in the model group. However, how endothelial injury exacerbates this pathological process through metabolic networks remains unclear. As an important tool in systems biology, metabolomics can comprehensively reveal the dynamic changes of small molecule metabolites in brain tissue. By comparing the differential metabolites among Model, WKY and SHR group, it can screen the key metabolic pathways mediated by endothelial injury.

**2.4.1. Expression and functional analysis of differential metabolites between Model group and WKY group.** Compared with Model and WKY, there were 437 significantly different metabolites, 128 (+) and 309 (-). The top 50 differential metabolites are all expressed (-) and clustered into 12 categories: phosphatidyl acid (n = 17), amino acids,

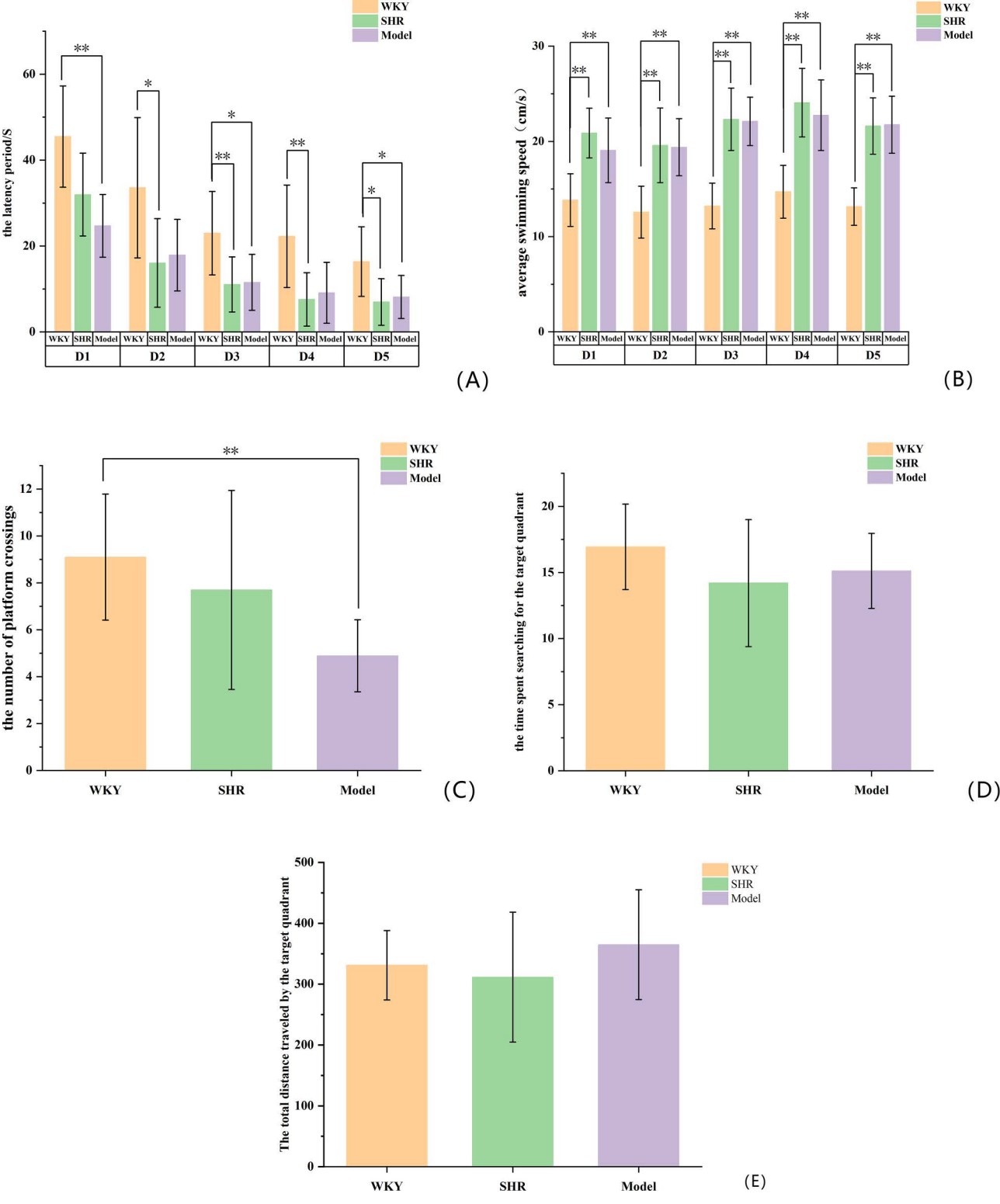

**Fig 1. (A) Escape latency; (B) Average swimming speed; (C) The number of platform crossings; (D) The time spent searching for the target quadrant; (E) The total distance traveled by the target quadrant.** *$P<0.05$ compared with the WKY group; **$P<0.01$ compared with the WKY group.

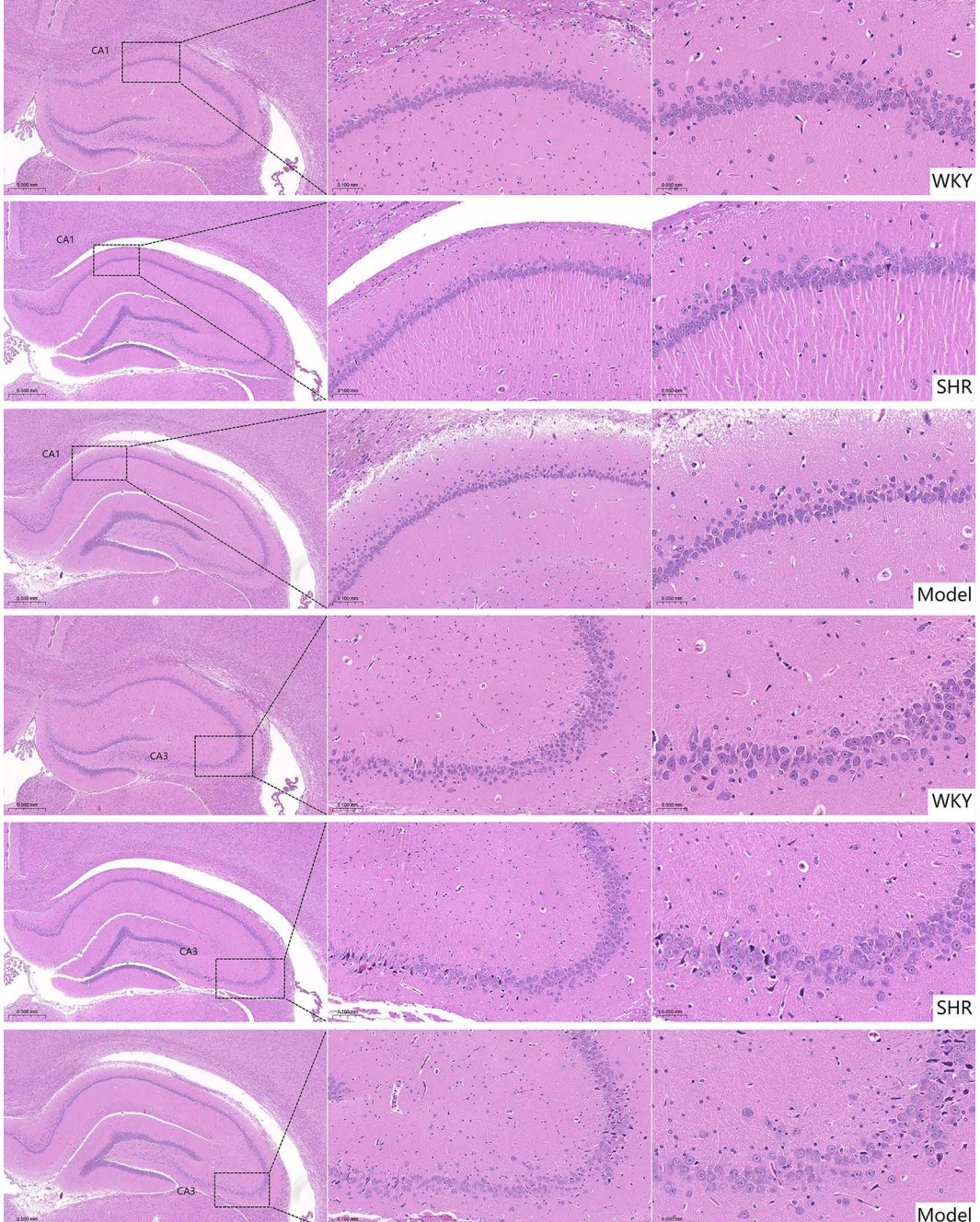

**Fig 2. Pathological observation of CA1 and CA3 regions of the hippocampus in each group (HE staining, ×200, bar: 100um; ×400, bar: 50um).**

peptides and analogs (n = 13), amines, choline, and other organic nitrogen compounds (n = 3), glycation end products (n = 3), nucleosides, nucleotides and analogs (n = 4), vitamins and their derivatives (n = 2), sphingolipids (n = 1), organic Acids and their derivatives (n = 1), indoles and other heterocyclic compounds (n = 2), fatty acids (n = 2), flavonoids, benzene and substituted derivatives (n = 1), acyl meat Base (n = 1). See Table 1 and Figs 4 A~B. Analyze the data flow direction of differential metabolites and KEGG pathways. The Top15 pathways are shown in Table 2 and Fig 4C.

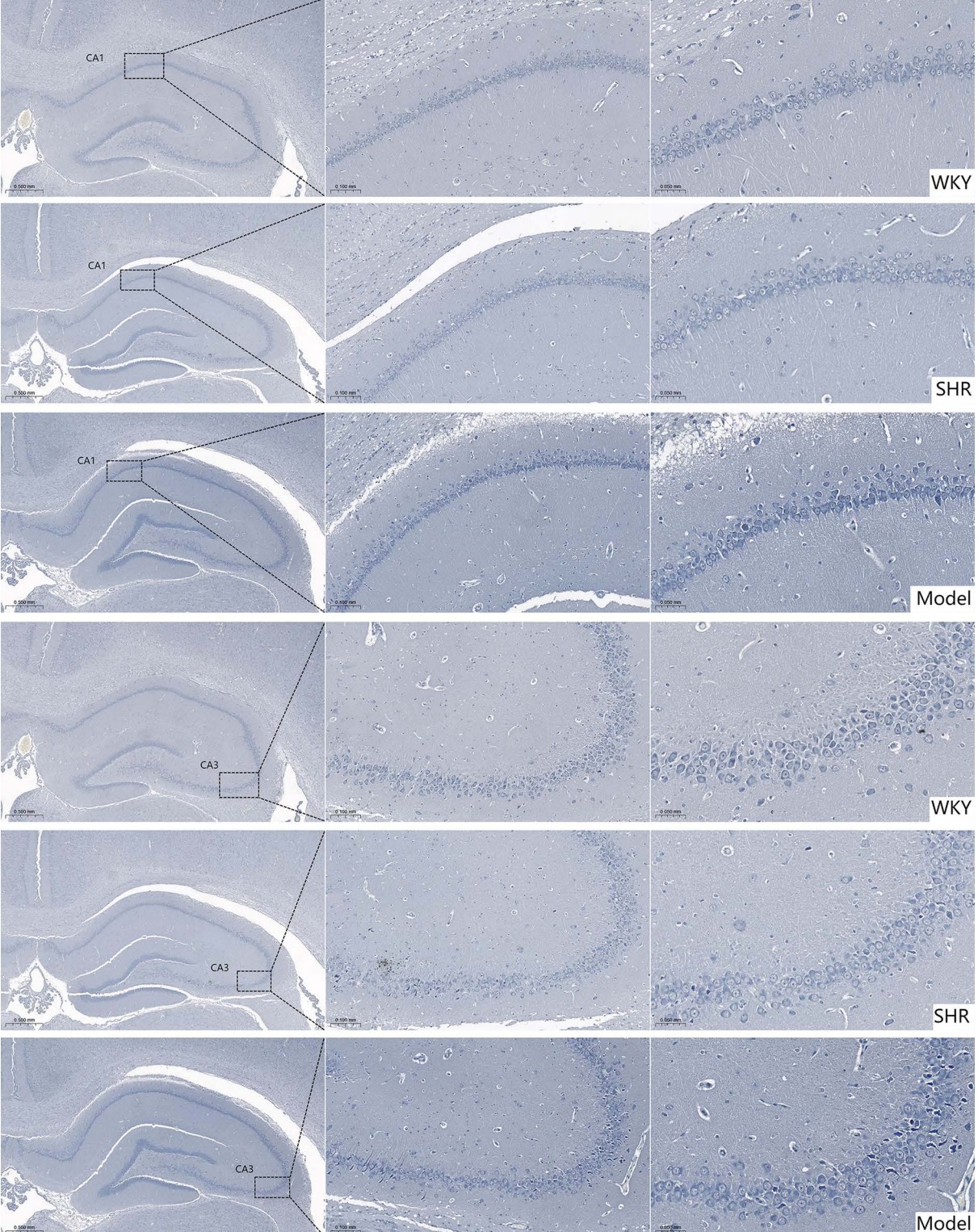

**Fig 3. Pathological observation of CA1 and CA3 regions of the hippocampus in each group (Nissl staining, × 200, bar: 100um; × 400, bar: 50um).**

**Table 1. OPLS-DA model parameter.**

| group | $R^2X(cum)$ | $R^2Y(cum)$ | $Q^2(cum)$ |
|---|---|---|---|
| Model group. Vs. WKY group | 0.668 | 1.000 | 0.994 |
| SHR group. Vs. WKY group | 0.400 | 0.994 | 0.859 |
| Model group. Vs. SHR group | 0.751 | 1.000 | 0.995 |

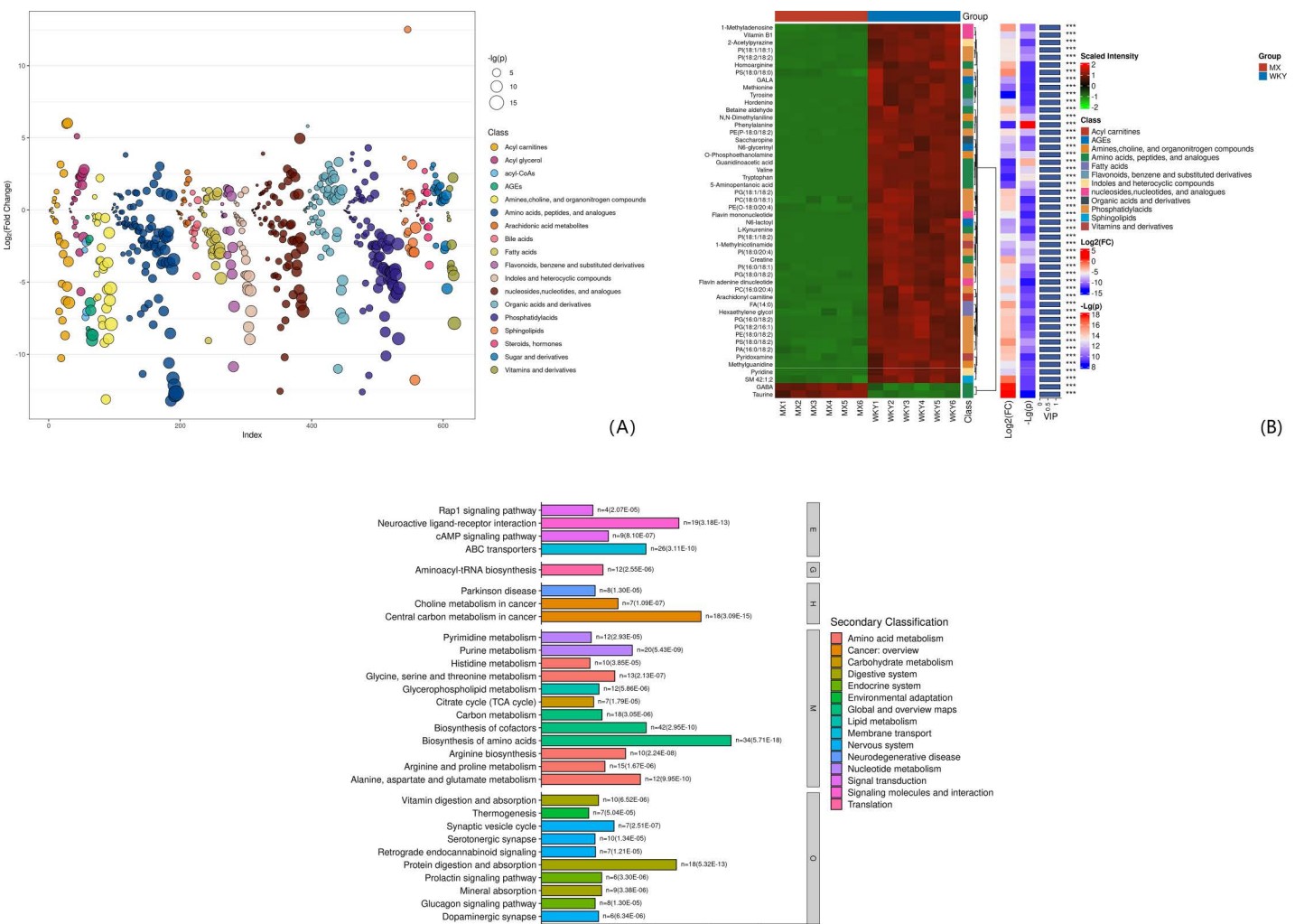

**Fig 4. (A)** FC and VIP bubble chart of Model.vs.WKY comparison group; **(B)** Hierarchical clustering results of differential metabolites in Model.vs.WKY comparison group (only the top 50 VIP values are plotted); **(C)** Bar chart of enriched pathways of top 30 differential metabolites in Model.vs.WKY comparison group (Level 1 pathway classification: Metabolism (M), Genetic Information Processing (G), Environmental Information Processing (E), Organic Systems (O), Human Diseases (H)).

**2.4.2. Expression and functional analysis of differential metabolites between Model group and SHR group.** Model and SHR ratio, 449 significantly different metabolites, 119 (+), 330 (-). The Top 50 differential metabolites are all expressed (-) and clustered into 11 categories: phosphatidyl acid (n = 17), amino acids, peptides and analogs (n = 8), nucleosides, nucleotides and analogs (n = 6), indoles and other heterocyclic compounds (n = 5), amines, choline,

**Table 2. Model vs. WKY group KEGG pathways (Top 15).**

| Description | pvalue | Count | UP | DOWN |
|---|---|---|---|---|
| ABC transporters | 3.10652E-10 | 26 | 6 | 20 |
| Alanine, aspartate and glutamate metabolism | 9.95371E-10 | 12 | 6 | 6 |
| Aminoacyl-tRNA biosynthesis | 2.54628E-06 | 12 | 1 | 11 |
| Arginine and proline metabolism | 1.67072E-06 | 15 | 2 | 13 |
| Arginine biosynthesis | 2.23768E-08 | 10 | 3 | 7 |
| Biosynthesis of amino acids | 5.71287E-18 | 34 | 10 | 24 |
| Biosynthesis of cofactors | 2.94919E-10 | 42 | 11 | 31 |
| cAMP signaling pathway | 8.09722E-07 | 9 | 1 | 8 |
| Central carbon metabolism in cancer | 3.0881E-15 | 18 | 7 | 11 |
| Choline metabolism in cancer | 1.08741E-07 | 7 | 1 | 6 |
| Glycine, serine and threonine metabolism | 2.13186E-07 | 13 | 4 | 9 |
| Neuroactive ligand-receptor interaction | 3.17544E-13 | 19 | 3 | 16 |
| Protein digestion and absorption | 5.32119E-13 | 18 | 2 | 16 |
| Purine metabolism | 5.42789E-09 | 20 | 7 | 13 |
| Synaptic vesicle cycle | 2.50875E-07 | 7 | 1 | 6 |

and other organic nitrogen compounds (n=3), fatty acids (n=3), glycation end products (n=2), flavonoids, benzene and substituted derivatives (n=2), sphingolipids (n=2), vitamins and their derivatives (n=1), organic acids and their derivatives (n=1). See Table 3 and Figs 5 A~B. Analyze the data flow direction of differential metabolites and KEGG pathways. The Top15 pathways are shown in Table 3 and Fig 5C.

### 2.4.3. Summary of differential metabolite expression in the WKY group, SHR group, and model group.
Compared with the WKY group and SHR group, the Model group exhibits a decrease in the levels of Alanine, Arginine, Aspartate, ATP, cAMP, Creatine, Cyanocobalamin, Dopamine, Guanidinoacetic acid, Indole, Serotonin, TMP, Tryptophan, Uric acid and Vitamin B1 ($P<0.01$). In contrast, GABA, Glucose, Isocitrate, Malate, Threonine and Pyrophosphate show a increasing trend ($P<0.01$) (Table 4).

**Table 3. Model vs. SHR group KEGG pathways (Top 15).**

| Description | pvalue | Count | UP | DOWN |
|---|---|---|---|---|
| ABC transporters | 1.99628E-10 | 26 | 6 | 20 |
| Alanine, aspartate and glutamate metabolism | 7.85782E-10 | 12 | 5 | 7 |
| Arginine and proline metabolism | 3.51685E-08 | 16 | 2 | 14 |
| Arginine biosynthesis | 1.83608E-08 | 10 | 3 | 7 |
| Biosynthesis of amino acids | 3.25756E-21 | 37 | 10 | 27 |
| Biosynthesis of cofactors | 1.15039E-11 | 44 | 10 | 34 |
| cAMP signaling pathway | 4.84679E-08 | 10 | 1 | 9 |
| Carbon metabolism | 1.00841E-07 | 20 | 13 | 7 |
| Central carbon metabolism in cancer | 2.16204E-15 | 18 | 6 | 12 |
| Choline metabolism in cancer | 2.12297E-09 | 8 | 1 | 7 |
| Dopaminergic synapse | 2.17561E-07 | 7 | 0 | 7 |
| Neuroactive ligand-receptor interaction | 3.30192E-13 | 19 | 2 | 17 |
| Nucleotide metabolism | 9.75555E-16 | 22 | 5 | 17 |
| Protein digestion and absorption | 3.75657E-13 | 18 | 2 | 16 |
| Synaptic vesicle cycle | 6.12459E-09 | 8 | 1 | 7 |

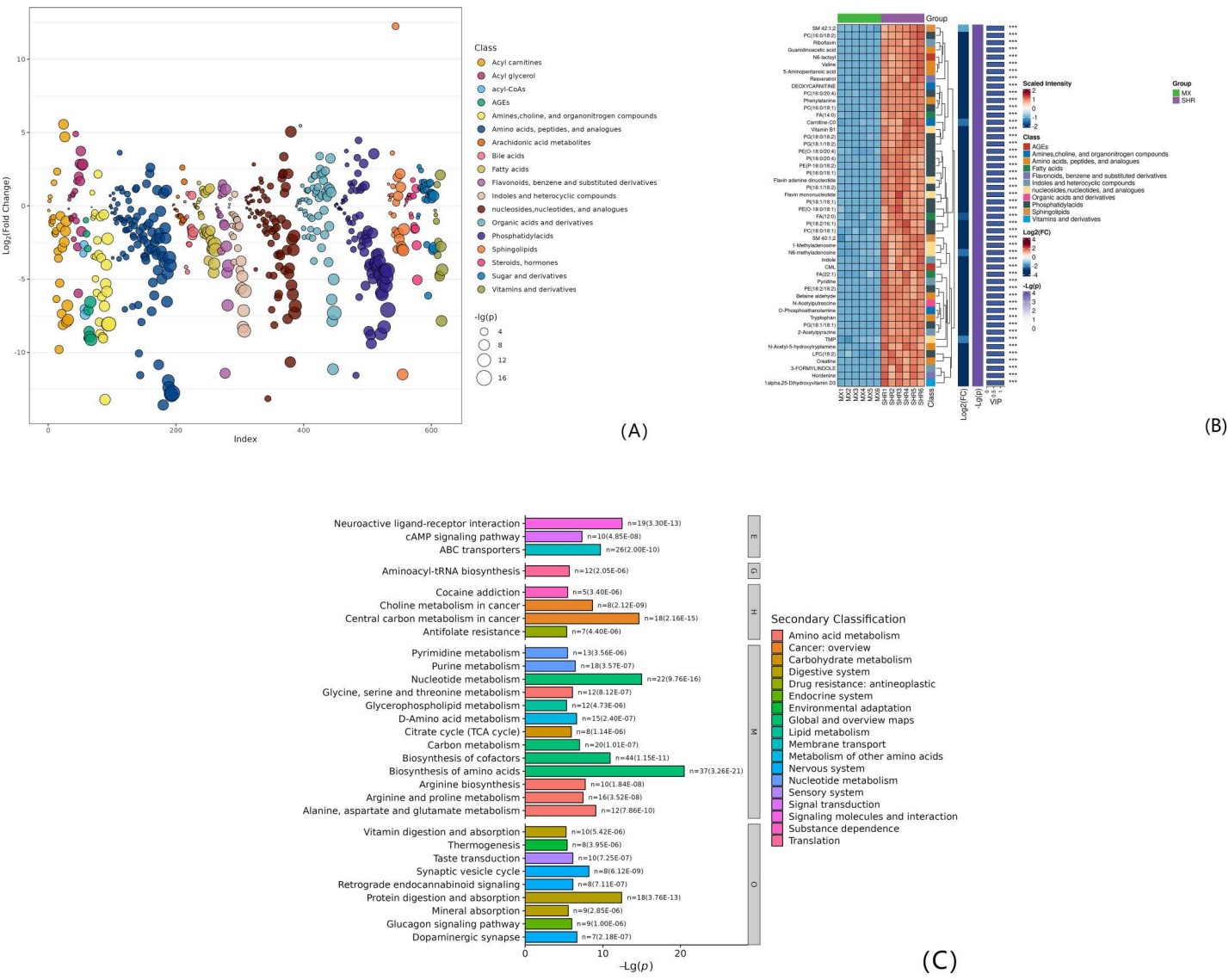

**Fig 5. (A)** FC and VIP bubble plots of the Model.vs.SHR; **(B)** Hierarchical clustering results of differential metabolites in the Model.vs.SHR (only the top 50 metabolites with the highest VIP values are plotted); **(C)** Bar chart of enriched pathways of the top 30 differential metabolites in the Model.vs.SHR.

## 2.5. Comparison of differential proteins in rat brain tissues in each group

Based on previous behavioral comparisons, observations of pathological staining of hippocampal tissues, and analysis of differential metabolites among rats in the WKY group, SHR group, and Model group, we further explore the molecular differences in brain tissues of the three groups of rats at the protein level. We identify differentially expressed proteins between the Model group and the WKY group, as well as between the SHR group and the Model group, and analyze the functions of these differential proteins and their enriched pathways. This aims to provide a protein-level basis for understanding related physiological or pathological processes, complement research results such as metabolomics, and improve the multi-dimensional understanding of differences among rats in each group.

**Table 4. The changing trends of the main differential metabolites in the WKY, SHR and Model groups.**

| Name | WKY | SHR | Model |
|------|-----|-----|-------|
| Alanine | – | – | down**ᐃᐃ |
| Arginine | – | – | down**ᐃᐃ |
| Aspartate | – | – | down**ᐃᐃ |
| ATP | – | – | down**ᐃᐃ |
| cAMP | – | – | down**ᐃᐃ |
| Creatine | – | – | down**ᐃᐃ |
| Cyanocobalamin | – | – | down**ᐃᐃ |
| Dopamine | – | – | down**ᐃᐃ |
| GABA | – | – | up**ᐃᐃ |
| Glucose | – | – | up**ᐃᐃ |
| Guanidinoacetic acid | – | up** | down**ᐃᐃ |
| Indole | – | up** | down**ᐃᐃ |
| Isocitrate | – | up** | up**ᐃᐃ |
| Malate | – | – | up**ᐃᐃ |
| Threonine | – | – | up**ᐃᐃ |
| Phenylalanine | – | up** | down**ᐃᐃ |
| Pyrophosphate | – | – | up**ᐃᐃ |
| Serotonin | – | – | down**ᐃᐃ |
| TMP | – | – | down**ᐃᐃ |
| Tryptophan | – | – | down**ᐃᐃ |
| Uric acid | – | – | down**ᐃᐃ |
| Vitamin B1 | – | up** | down**ᐃᐃ |

Compared with WKY, $*P < 0.05$, $**P < 0.01$; Compared with SHR, $^{△}P < 0.05$, $^{△△}P < 0.01$.

**2.5.1. Expression and functional analysis of differential proteins between Model group and WKY group.** Compared with the WKY group, there were 141 differentially expressed proteins in the Model group, of which 55 were up-regulated and 86 were down-regulated, see Fig 6A. The differential proteins between the Model group and the WKY group are enriched in 87 pathways. According to the level 1 level, 30 (34%) are enriched in human diseases, 19 (22%) in biological systems, 18 (21%) in metabolism, and environmental information. There are 12 processes (14%), 4 cellular processes (5%), 3 genetic information processes (3%), and 1 pathway has no attribution, see Fig 6B. A network of significantly enriched pathways and protein/protein coding genes and protein interactions in the pathway was constructed, including a total of 32 proteins, 12 (+), and 20 expressed (-), as shown in Fig 6C.

**2.5.2. Expression and functional analysis of differential proteins between SHR group and Model group.** Compared with the SHR group, there were 28 differentially expressed proteins in the Model group, of which 15 were up-regulated and 13 were down-regulated, see Fig 7A. Differential proteins between the SHR group and the Model group are enriched in 24 pathways. According to the level 1 level, 9 pathways (38%) are enriched in human diseases, 6 pathways (25%) in biological systems, 3 pathways (13%) in metabolism, and environmental information. There are 3 processing items (13%), 2 cellular process items (8%), and 1 genetic information processing item (4%), see Fig 7B. A network of significantly enriched pathways and protein/protein coding genes and protein interactions in the pathway was constructed. There were 5 proteins in total, 4 expressed (+) and 1 expressed (-), as shown in Fig 7C.

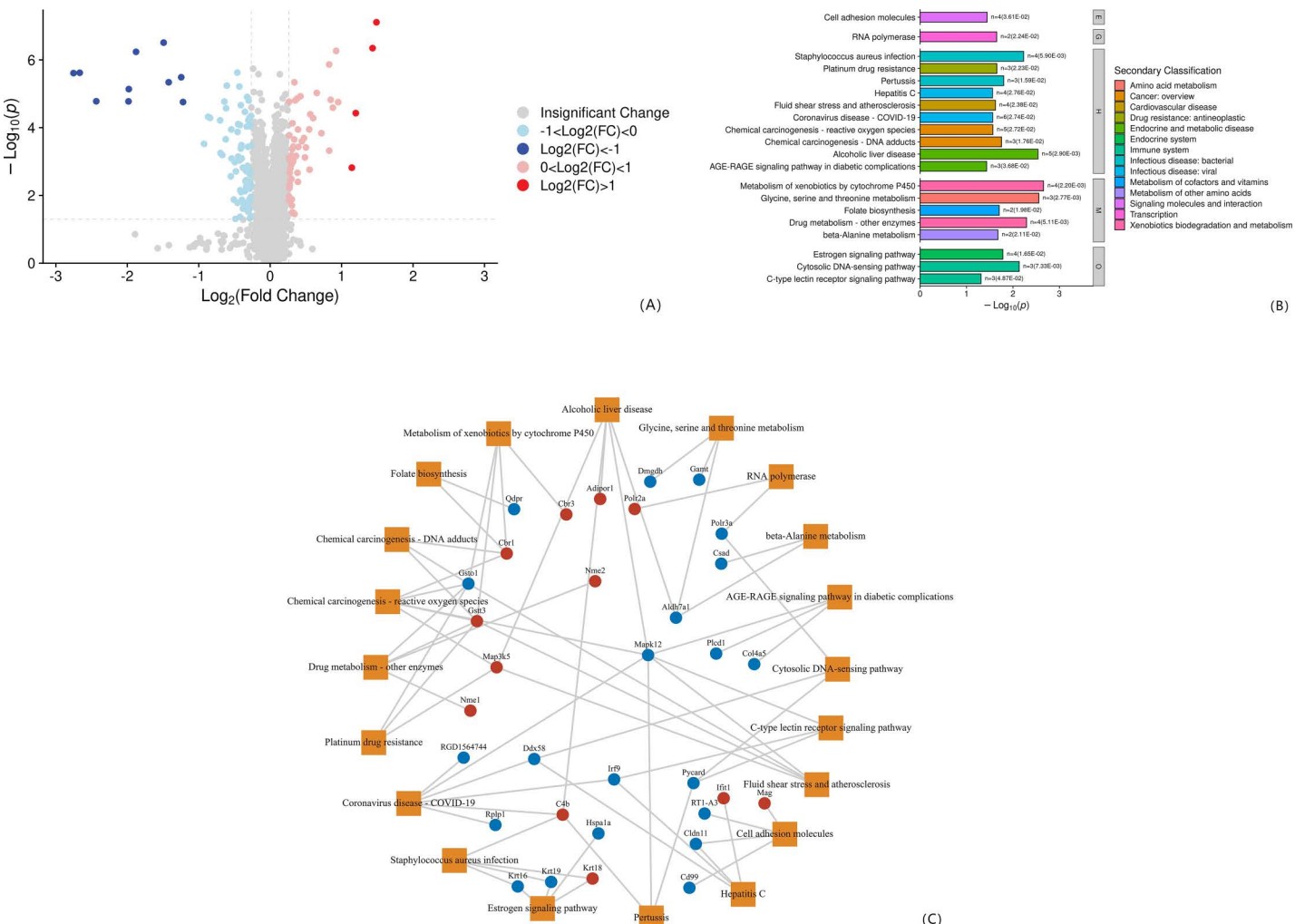

**Fig 6. (A)** Hierarchical clustering results of differential proteins in the Model.vs.WKY; **(B)** Differential protein enrichment bar chart in the Model.vs.WKY; **(C)** Significantly enriched pathways and those in the pathway Protein/protein coding genes and protein interaction network.

## 3. Discussion

Vascular cognitive impairment is caused by damage to vascular units due to multiple risk factors (hypertension, hyper-lipidemia, hyperglycemia, etc.), which ultimately leads to VD [9]. Therefore, the VCI (Vascular Cognitive Impairment) model, in which thrombosis and blockage are formed due to micro-blood flow stimulation, resulting in cerebral ischemia and hypoxia, can simulate the clinical onset mode and pathophysiological characteristics. In the past, vascular occlusion, intravascular embolism and spontaneous VCI models have been widely used in scientific research. Washida [10] found that asymmetric ligation of bilateral common carotid arteries (the right common carotid artery was implanted with an amilo constrictor to gradually occlude the blood vessels within 28 days, and the left side was implanted with a microcoil to cause about 50% arterial stenosis) can cause chronic cerebral hypoperfusion, cerebral white matter infarction and gliosis in rats. Xue [11] proved that after injection of microsphere vascular embolic agents into the internal carotid artery of SD rats, cerebral blood flow was reduced, escape latency was shortened and the number of perforations was reduced. The spontaneously hypertensive rat (SHR) is an animal model with chronic hypertension and persistent susceptibility to ischemia/

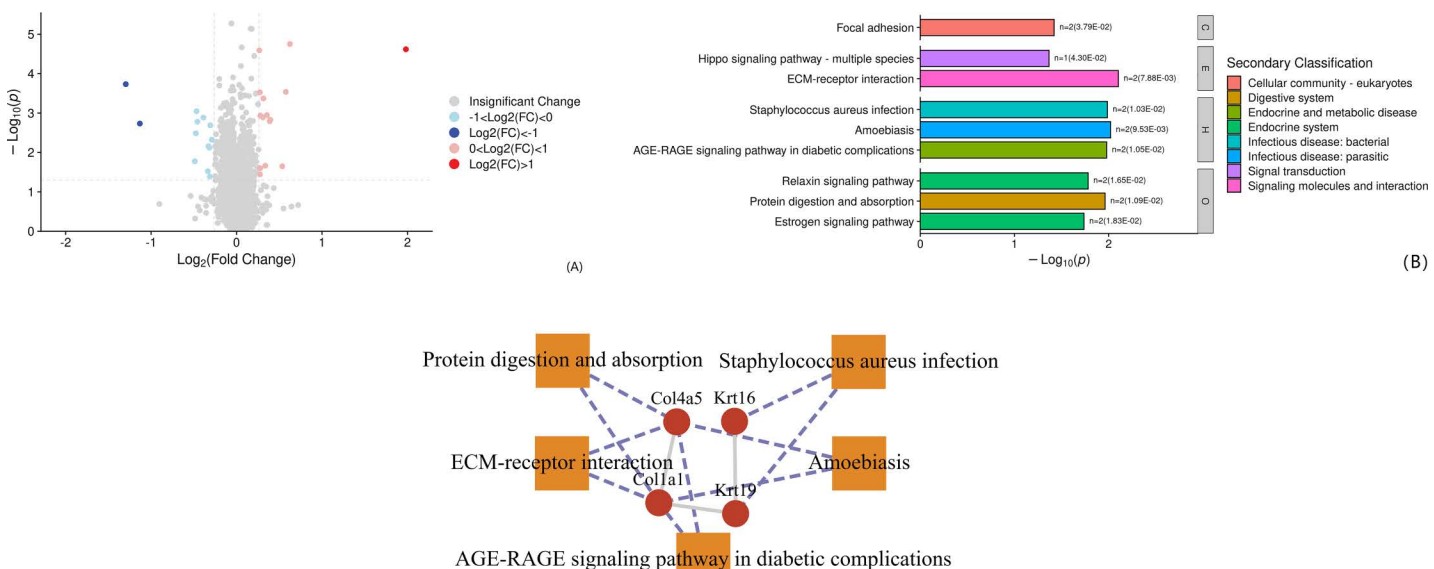

**Fig 7. (A) Hierarchical clustering results of differential proteins in SHR vs Model; (B) Differential protein enrichment bar chart in SHR vs Model; (C) Significantly enriched pathways and proteins/proteins in the pathway Coding gene and protein interaction network.**

hypoxia injury. Previous studies have found that in SHR rats, the gray matter volume and immune response areas in the CA1 sub-region and the dentate gyrus are reduced, the expression of the cytoskeletal component, neurofilament protein, is decreased, and astrogliosis occurs [12]. The above findings indicate that the brain tissue of SHR rats has undergone degenerative changes and developed an astrocyte reaction similar to that of VCI. However, simply reducing cerebral perfusion or simply imitating risk factors and waiting for it to spontaneously occur cannot fully simulate the complex pathological process of VCI. Therefore, the research team used the SHR + microcurrent stimulation of bilateral common carotid arteries to induce endothelial injury and thrombosis modeling method, combined with the risk factors and pathogenesis of VCI to replicate a comprehensive animal model.

The main manifestation of VCI is the slow and gradual decline of mental functions including memory, judgment and learning ability. Behavioral comparisons between rats in each group found that from D1 to D5, compared with WKY, the latency of the Model group (SHR+vascular injury composite model) and SHR group (no vascular injury) was shorter, the average swimming speed increased, and there was statistical significance ($P < 0.05$, $P < 0.01$). This may be related to the type of rat itself. WKY rats showed behavioral defects in multiple behavioral tests, especially in long-term darkness. Under the environment, their autonomous activities were significantly weakened, and in comparison, SHR rats had stronger activity abilities. On D6, the platform was removed and the spatial exploration test was started. Compared with the WKY group, the number of platform crossings in the Model group (SHR+vascular injury composite model) was significantly reduced and the search time for the target quadrant were reduced ($P < 0.05$), which suggested the spatial learning and memory ability of the rats in the model group (SHR+vascular injury composite model). Injury, SHR + microcurrent stimulation of bilateral common carotid arteries can simulate the clinical characteristics of cognitive dysfunction in patients with VCI. However, there is no statistical significance in the behavioral comparison between the SHR group (no vascular injury) and the Model group (SHR+vascular injury composite model). SHR rats are similar to human essential hypertension, with normal blood pressure at birth and gradually developing into stable blood pressure within the first 2–4 months after birth. of hypertension [13]. In 4-month-old SHR rats, a limited reduction in white matter volume can be observed. In 6-month-old SHR rats, the volume of hippocampal CA1 region and dentate gyrus gray matter is reduced, the expression

of glial fibrillary acid protein is increased, and the number of astrocytes is increased [13]. In this study, SHR group (no vascular injury) aged 13–14 weeks were selected, and the samples were collected 1 week after modeling, which shows that SHR rats (no vascular injury) can already show similar manifestations of VCI. Therefore, compared with the WKY group, the number of platform crossings, target quadrant search time and distance in the SHR group (no vascular injury) were reduced ($P > 0.05$). When comparing the Model group (SHR+vascular injury composite model) with the WKY group, the number of platform crossings in the Model group (SHR+vascular injury composite model) was decreased ($P < 0.01$), while there was no statistical significance when comparing the SHR group (no vascular injury) with the Model group (SHR+vascular injury composite model). HE staining was used to observe the pathological state of brain tissue in rats in each group, and Nissl staining was used to observe the state of neuronal Nissl bodies. The results of HE staining showed that compared with the WKY group, the number of neurons in the hippocampal tissue structure of rats in the SHR group (no vascular injury) and the Model group (SHR+vascular injury composite model) was reduced, the nuclei were deformed and condensed, the staining was deepened, and the cell bodies were relatively disordered. The results of Nissl staining showed that the hippocampal neurons in the SHR group (no vascular injury) and the Model group (SHR+vascular injury composite model) were irregularly arranged and sparsely distributed, with lighter Nissl staining, and the morphology of some hippocampal neurons in the Model group (SHR+vascular injury composite model) changed from granular to powdery, with pyknosis of the cell nucleus, accompanied by cell edema and necrosis, which indicated that the model was successfully established.

VCI and VD have been found to be associated with metabolic changes [14]. Further elucidation of the pathophysiological characteristics of VCI will help reduce its high incidence. In this study, we used a method combining TMT quantitative proteomics technology with liquid chromatography tandem mass spectrometry (LC-MS/MS) to analyze metabolomics and proteomics data in brain tissue samples to reveal changes in metabolites and proteins in VCI. Based on a multi-omics joint analysis method, a reliable network for the systematic biological analysis of VCI from proteins to final metabolites was constructed.

Amino acid metabolism disorder is one of the factors that lead to the formation of VD [14]. Tryptophan intervenes in the function of the central nervous system by regulating indole derivatives, kynurenine pathways and serotonin synthesis. Its concentration is related to the onset of chronic cerebral ischemia, neuronal damage and neurodegenerative diseases [15,16]. Guanidinoacetic acid is a precursor of creatine, creatine can increase the energy supply of brain cells, especially in the form of creatine phosphate, maintain the ATP level of brain cells and enhance memory by increasing the synthesis of neurotransmitters such as acetylcholine and reducing oxidative stress damage to brain cells [17]. Metabolomics results showed that compared with the SHR group (no vascular injury) and the WKY group, the expression of tryptophan, guanidinoacetic acid and creatine in the Model group (SHR+vascular injury composite model) was downregulated, which is similar to the results of previous studies. Excessive activation of excitatory amino acids is considered an important link in neuronal damage [18,19]. Compared with the control group, the differential metabolites in the Model group (SHR+vascular injury composite model) were mainly concentrated in amino acid metabolism pathways such as alanine, aspartic acid and glutamic acid metabolism, which Aspartate and glutamate are both excitatory amino acids. The proteomics differential proteins were also enriched in amino acid metabolism, confirming the metabolomics results.

In addition, compared with the SHR group (no vascular injury), the contents of thiamine monophosphate (TMP) and vitamin B1 in the Model group (SHR+vascular injury composite model) decreased, while the content of glucose increased, which is similar to previous studies [20]. TMP is an intermediate form in the process of vitamin $B_1$ being converted to thiamine pyrophosphate (TPP) in the body. TPP is a coenzyme of key enzyme systems such as the α-ketoacid oxidative decarboxylase multienzyme complex and is essential for glucose metabolism. The reduction of TMP and TDP may lead to reduced glucose metabolism in brain tissue, the formation of neuritic plaques and hyperphosphorylation of tau protein, and promote the occurrence of cognitive dysfunction [20,21]. A clinical study has shown that the level of vitamin $B_{12}$ is positively correlated with the Montreal Cognitive Assessment. That is to say, a decrease in the level of vitamin $B_{12}$ may

lead to cognitive dysfunction and promote the occurrence of VD [22]. Compared with the WKY group and the SHR group (no vascular injury), the level of vitamin $B_{12}$ in the Model group (SHR+vascular injury composite model) decreased, which is consistent with previous studies.

Indices related to neurotransmitter metabolism and oxidative stress also showed significant changes in the Model group (SHR+vascular injury composite model). Tryptophan is a precursor for serotonin synthesis, dopamine, serotonin and tryptophan play crucial roles in regulating emotions, cognition and behavior [23]. Uric acid is the end product of purine metabolism in the body, and its increase may be related to enhanced oxidative stress [24]. During the pathological process of vascular dementia, oxidative stress can lead to damage to neurons and blood vessels, trigger neuroinflammation, further disrupt the balance of the neurotransmitter system, and exacerbate cognitive function impairment [25]. This study found that the levels of dopamine and serotonin in the Model group (SHR+vascular injury composite model) decreased significantly, while the level of uric acid increased significantly. In the Model group (SHR+vascular injury composite model), the level of γ-aminobutyric acid (GABA) increased significantly. Under pathological conditions such as vascular dementia, abnormal activation of the GABAergic system may lead to excessive inhibition of neuronal activity, affect the transmission and integration of neural signals, and further impair cognitive function. [26].

Relaxin can affect brain memory by inhibiting the synthesis of cAMP and reducing the activity level of the cAMP-PKA pathway [27]. It can also reduce brain inflammation to improve white matter loss, reduced cortical thickness and cognitive dysfunction [28]. Focal adhesions are physical contact points between the extracellular matrix and the cell actin cytoskeleton [29]. In the nervous system, the connection and adhesion between neurons are crucial for the transmission of information and the maintenance of cognitive function [30]. Impaired function of focal adhesions may affect the normal connection and communication of neurons, thereby having a negative impact on cognitive function. Previous metabolomics studies have also suggested that focal adhesions play a role in cognitive dysfunction [31]. In this study, the differentially expressed proteins between the SHR group (no vascular injury) and the Model group (SHR+vascular injury composite model) were enriched in the "Relaxin Signaling Pathway" and "Focal Adhesion" pathways, which also confirms that relaxin and focal adhesions have a certain influence on the development of hypertension to vascular dementia.

In summary, the VCI rat model caused by hypertension combined with endothelial injury established in this study simulates the clinical characteristics of cognitive dysfunction caused by risk factors of VCI and is an ideal animal model. Compared with the spontaneous SHR model, the SHR+microcurrent stimulation of bilateral common carotid arteries to cause endothelial injury and thrombosis can simulate vascular injury under hypertension, thereby causing changes in cerebral hemodynamics and hypoxia, ischemia and other damage to brain tissue, which are important pathological mechanisms of VCI. Therefore, this animal model is closer to the pathological process of human VCI and enables the model to show VCI and even characteristic pathological changes of VD more quickly. In addition, the application of metabolomics and proteomics has important guiding value in studying the diagnostic indicators and therapeutic targets of VCI and VD patients. VCI and VD mediated by hypertension as a risk factor mainly involve pathways such as amino acid metabolism, relaxin signaling pathway and focal adhesion. Metabolites such as TMP and vitamin B1 also play a certain role in the occurrence and development of VD. Although there are differences between animal models and VCI patients, this analysis also shows a high degree of consistency with the information of VCI and VD patients [32]. This study analyzed valuable biomarkers and related metabolic pathways, which can also provide a reference for the pathogenesis of VCI and VD. This study still has the following defects: Firstly, the scope of histological assessment is limited: it focuses only on the pathological changes of the hippocampus and fails to assess other cognitive-related brain regions such as the prefrontal cortex and amygdala. Clinical patients with VCI are often accompanied by multi-brain region damage, and the analysis of a single brain region may underestimate the overall extent of brain damage. In the future, it is necessary to expand the scope of histological assessment to improve the elucidation of pathological mechanisms..Secondly, only the Morris Water Maze was used to evaluate spatial learning and memory abilities, and behavioral tests related to depression or emotional domains (such as the Forced Swimming Test and Sucrose Preference Test) were not included. The missing emotional

assessment dimension in this study limits the authenticity of the model in simulating clinical VCI. Thirdly, the research of the research team is still preliminary. In future studies, molecular biological methods should be used to verify the signal transduction pathways, and combined with human samples (such as serum) of patients for verification, so as to screen out accurate and efficient biomarkers.

## Conclusion

Rats with hypertension combined with endothelial injury have behavioral disorders, which are similar to the causes and signs of clinical vascular cognitive impairment; the biological mechanisms manifested in the model involve amino acid metabolism disorders, energy metabolism, relaxin signaling pathway and impaired focal adhesion function. Characteristic metabolites in the model group (SHR+vascular injury composite model), such as ATP, cAMP, Creatine, Cyanocobalamin, Dopamine, Serotonin, Tryptophan, Uric acid and Vitamin $B_1$ decreased, while GABA, Glucose, Isocitrate and Malate increased.

## Supporting information

**S1. The abundance values of metabolites in the Model group and the WKY group.**
(XLSX)

**S2. The abundance values of metabolites in the Model group and the SHR group.**
(XLSX)

**S3. Differentially expressed protein in the Model group and the WKY group.**
(XLSX)

**S4. Differentially expressed protein in the Model group and the SHR group.**
(XLSX)

**S5. Behavioral data.**
(XLSX)

## Acknowledgments

The combined analysis of metabolomics and proteomics in this project was assisted by Engineer Du Dan of Shanghai Baipu Biotechnology Co., Ltd., and we would like to express our sincere gratitude!

## Author contributions

**Conceptualization:** Surui Chang, Jiangang Liu, Dongling Wang.

**Formal analysis:** Xiao Zhang, Jiangang Liu.

**Funding acquisition:** Jiangang Liu.

**Investigation:** Dongling Wang.

**Methodology:** Xiao Zhang, Surui Chang.

**Project administration:** Jiangang Liu.

**Software:** Xiao Zhang, Surui Chang.

**Writing – original draft:** Xiao Zhang, Surui Chang.

**Writing – review & editing:** Jiangang Liu.

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
