## [Decision Letter · Decision Letter 0]

9 Mar 2025

Dear Dr. Liu,

Thank you for submitting your manuscript to PLOS ONE. After careful consideration, we feel that it has merit but does not fully meet PLOS ONE’s publication criteria as it currently stands. Therefore, we invite you to submit a revised version of the manuscript that addresses the points raised during the review process.

We look forward to receiving your revised manuscript.

Kind regards,

Nafisa M. Jadavji, PhD, MSc, BSc

Academic Editor

PLOS ONE

Journal Requirements:

2. To comply with PLOS ONE submissions requirements, in your Methods section, please provide additional information regarding the experiments involving animals and ensure you have included details on (1) methods of sacrifice, and (2) efforts to alleviate suffering.

3. Thank you for stating the following financial disclosure: This study was supported by Major Increase and Decrease Projects at Central Level (No.2060302).  

Additional Editor Comments:

Dear Author,

The reviewers have listed several concerns about your submission. If you choose to resubmit, I urge to address each concern in your revised manuscript and respond.

Sincerely,

Nafisa

Reviewers' comments:

Reviewer's Responses to Questions

**Comments to the Author**

1. Is the manuscript technically sound, and do the data support the conclusions?

Reviewer #1: Partly

Reviewer #2: No

2. Has the statistical analysis been performed appropriately and rigorously?

Reviewer #1: No

Reviewer #2: No

3. Have the authors made all data underlying the findings in their manuscript fully available?

Reviewer #1: No

Reviewer #2: No

4. Is the manuscript presented in an intelligible fashion and written in standard English?

Reviewer #1: No

Reviewer #2: No

Reviewer #1: This is a good study. Provided the author addresses all the gaps and clarifies certain information in this manuscript:

Introduction:

In general, it's acceptable.

Methodology

The language in this submission requires improvement, as some sentences are unclear or awkwardly phrased. Additionally, the sentence arrangement could be better organised for coherence and readability. In the statistical analysis section, the explanation of the validity for OPLS-DA was missing, as were the values of the selected VIP and the criteria used for comparisons. Information about the software used for statistical analysis was also not provided.While comparisons between groups (intergroup comparisons) were mentioned, it is unclear whether intragroup comparisons were performed. This should be clarified. No references were provided for some of the major methods.

The total number of rats was stated as 30, with 15 SHR and 15 WKY. However, in Section 1.3, it is mentioned that there are three groups: WKY (10 rats), model (10 rats), and SHR (10 rats). Does the model group comprise 5 SHR and 5 WKY?

Results and discussion

More discussion is needed and references. It is suggested to provide a diagram to highlight which metabolites are shared and different among those three groups.

Conclusion:

Didn't highlight the major findings in the study—too simple and general.

Others:

Kindly provide the limitations of the study. Formatting was not consistent throughout the manuscript (example: 1.5.2).

Please find the comments in the attached documents too.

Reviewer #2: The paper studies the molecule mechanism of vascular cognitive impairment using model rats by various methods, such as metabolomics, proteomics, Morris water maze and pathological slides. The paper is not well written in good language, which makes it difficult to understand. Moreover, the statistical and computational analysis were not clearly described, and the results are not well analyzed. Please see the following comments:

Section 1.8. P<0.05 threshold should be adjusted because of the multiple testing problem to reduce false discovery rate.

Figure 2. What does the error bar mean? If the error bars mean confidence interval, since many of the overlap with each other, the difference between different groups should not be significant. Hence, the significant levels (***, **) are questionable.

What softwares were used to annotate the metabolites? The authors should report the metabolite annotation level of the analysis.

What softwares were used to annotate protein?

Figure 5A and 6A. what is the total number of metabolites shown? The abundance table of them should be provided as supplementary material. It seems there are more than 437 and 449 significant metabolites.

Figure 5C and 6C. They look very confusing. Please describe more.

There are a lot of typo and grammar issues. e.g.

- Section 1.1 Is there is typo here? 15 -> 15 rats?

- page 7 last paragraph. "After the operation ...". Please rewrite the sentence with correct grammar

- Section 1.8. "Univariate statistical analysis ...". Please rewrite this sentence with correct grammar

**Do you want your identity to be public for this peer review?** For information about this choice, including consent withdrawal, please see our Privacy Policy

Reviewer #1: No

Reviewer #2: No

---

## [Author Response · Author response to Decision Letter 1]

25 Apr 2025

We would like to express our sincere gratitude to the editor and the reviewers for their conscientiousness and responsibility. They promptly pointed out the issues in our article and kindly provided us with an opportunity for revision. We have polished the language of the article and made modifications to the images and layout. We sincerely hope that these revisions will meet your satisfaction.

Journal Requirements:

Response: The manuscript has been revised in accordance with the formatting requirements of PLOS ONE.

2. To comply with PLOS ONE submissions requirements, in your Methods section, please provide additional information regarding the experiments involving animals and ensure you have included details on (1) methods of sacrifice, and (2) efforts to alleviate suffering.

Response: During the rat modeling process, sodium pentobarbital was administered via intraperitoneal injection for anesthesia to alleviate the suffering of the rats. Euthanasia of the rats was performed by intravenous injection of an excessive dose of sodium pentobarbital. The relevant descriptions have been added to the article.

3. Thank you for stating the following financial disclosure: This study was supported by Major Increase and Decrease Projects at Central Level (No.2060302). Please state what role the funders took in the study. If the funders had no role, please state: ""The funders had no role in study design, data collection and analysis, decision to publish, or preparation of the manuscript.""

Response: Relevant descriptions have been added both in the article and the cover letter.

Response: The ethical statement outside the Methods section of the manuscript has been deleted.

Response: The Supporting Information has been supplemented.

Reviewer #1:

1.The language in this submission requires improvement, as some sentences are unclear or awkwardly phrased. Additionally, the sentence arrangement could be better organised for coherence and readability. In the statistical analysis section, the explanation of the validity for OPLS-DA was missing, as were the values of the selected VIP and the criteria used for comparisons. Information about the software used for statistical analysis was also not provided.While comparisons between groups (intergroup comparisons) were mentioned, it is unclear whether intragroup comparisons were performed. This should be clarified. No references were provided for some of the major methods.

Response: The language of the manuscript has been improved. In the statistical section, an explanation of OPLS-DA and the comparison criteria for VIP values have been supplemented. This article conducts comparisons between groups and does not perform comparisons within groups. Relevant references have been added to the methodology section.

2.The total number of rats was stated as 30, with 15 SHR and 15 WKY. However, in Section 1.3, it is mentioned that there are three groups: WKY (10 rats), model (10 rats), and SHR (10 rats). Does the model group comprise 5 SHR and 5 WKY?

Response: The total number of rats is 30, which should be 10 in the WKY group, 10 in the Model group (that is, SHR + microcurrent stimulation group), and 10 in the SHR group. There was an error in the previous description, and thank you for pointing it out.

3.Results and discussion: More discussion is needed and references. It is suggested to provide a diagram to highlight which metabolites are shared and different among those three groups.

Response: The discussion section has been refined, and table related to the main differential metabolites among the three groups have been added.

4.Conclusion: Didn't highlight the major findings in the study—too simple and general.

Response: The conclusion section has been improved.

5.Others: Kindly provide the limitations of the study. Formatting was not consistent throughout the manuscript (example: 1.5.2).

Response: The research limitations have been added to the discussion section, and the article format has been corrected and revised.

Reviewer #2:

1.Section 1.8. P<0.05 threshold should be adjusted because of the multiple testing problem to reduce false discovery rate.

Response: For multiple testing, the Bonferroni and Dunnett T3 tests were selected. The P-values obtained from both methods are corrected values.

2.Figure 2. What does the error bar mean? If the error bars mean confidence interval, since many of the overlap with each other, the difference between different groups should not be significant. Hence, the significant levels (***, **) are questionable.

Response: The error bars specifically represent the mean±standard deviation. Other types of charts have been used to make the presentation clearer.

3.What softwares were used to annotate the metabolites? The authors should report the metabolite annotation level of the analysis.

Response: MT1000 Metabolite Identification Software: MultiQuant. It utilizes ion pairs for metabolite identification, and the identification levels are Level 1 and Level 2. It has been indicated in the text.

4.What softwares were used to annotate protein?

Response: Protein Database Search Software for TMT Project: Proteome Discoverer.

5.Figure 5A and 6A. what is the total number of metabolites shown? The abundance table of them should be provided as supplementary material. It seems there are more than 437 and 449 significant metabolites.

Response: The table of abundance of differential metabolites has been supplemented and uploaded.

6.Figure 5C and 6C. They look very confusing. Please describe more.

Response: The chart types of Figure 5C and Figure 6C have been changed to make the pathways clearer.

7.There are a lot of typo and grammar issues. e.g.

Response: The grammar and spelling of the article have been corrected and improved.

---

## [Decision Letter · Decision Letter 1]

9 Jul 2025

Dear Dr. Liu,

**We thank-you for making revisions to your manuscript and hope to read the revised manuscript soon. **

https://journals.plos.org/plosone/s/submission-guidelines#loc-laboratory-protocols . Additionally, PLOS ONE offers an option for publishing peer-reviewed Lab Protocol articles, which describe protocols hosted on protocols.io. Read more information on sharing protocols at https://plos.org/protocols?utm_medium=editorial-email&utm_source=authorletters&utm_campaign=protocols .

We look forward to receiving your revised manuscript.

Kind regards,

Nafisa M. Jadavji, PhD, MSc, BSc

Academic Editor

PLOS ONE

**Journal Requirements:**

**Additional Editor Comments:**

Dear Authors,

Thank-you for taking the time to make revisions to your manuscript. Some minor revisions are still required. We look forward to reading your revised manuscript.

Sincerely,

Nafisa

Reviewers' comments:

Reviewer's Responses to Questions

**Comments to the Author**

Reviewer #3: All comments have been addressed

Reviewer #4: (No Response)

Reviewer #5: All comments have been addressed

2. Is the manuscript technically sound, and do the data support the conclusions?

Reviewer #3: Yes

Reviewer #4: Yes

Reviewer #5: Yes

3. Has the statistical analysis been performed appropriately and rigorously?

Reviewer #3: Yes

Reviewer #4: Yes

Reviewer #5: Yes

4. Have the authors made all data underlying the findings in their manuscript fully available?

Reviewer #3: Yes

Reviewer #4: No

Reviewer #5: Yes

5. Is the manuscript presented in an intelligible fashion and written in standard English?

Reviewer #3: Yes

Reviewer #4: Yes

Reviewer #5: Yes

**Reviewer #3:**  The revised manuscript is well-structured, and the methodology and results are clearly presented. No major corrections appear necessary at this stage.

However, I would like to raise one conceptual point for clarification and potential revision. The authors have focused on pathological changes in the hippocampus, but vascular cognitive impairment (VCI) differs fundamentally from Alzheimer’s disease in its pathophysiology. VCI is typically associated with accumulated asymptomatic ischemic events, leading to white matter degeneration and diffuse global brain decline, rather than localized hippocampal dysfunction.

Therefore, while it is acceptable that hippocampal pathology was examined in this study, it would be helpful for the authors to clarify that:

Only the hippocampus was assessed histologically,

The behavioral evaluation mainly reflects cognitive performance and does not encompass depressive or affective domains,

The findings should be interpreted within these limitations.

Moreover, while the metabolomic and proteomic analyses are well conducted, many of the identified molecular changes—particularly those involving neurotransmitters and amino acid metabolism—are broadly linked to aging or psychiatric disorders. Some brief discussion on this point would strengthen the clinical relevance of the findings.

In summary, I recommend only minor textual revisions, particularly to the limitations and discussion sections, with the goal of making the implications more clinically accessible for practicing physicians. Overall, the manuscript is of high quality and should be considered favorably for publication after these small adjustments.

**Reviewer #4:**  PONE-D-24-59603R1

Combined metabolomics and proteomics analysis of vascular cognitive impairment in

hypertensive rats induced by endothelial injury

This is the first time I have seen this manuscript, but the paper was previously revised with provided. The authors have made many previous changes to text and methods sections.

The authors have looked at biological mechanisms of vascular cognitive impairment in SHR rat, following microcurrent damage, using metabolomics and proteomics tools as well as behaviour. The WKY rats and a “model” group serve as controls. The authors do notice changes in behaviour, which they relate suggest make the model clinically relevant, and they go on to describe various protein and metabolite pathways that appear different these group. The manuscript is a reasonable, descriptive study on a model vascular cognitive impairment.

Comments

While the authors studied both SHR, WHY and a model group and obtain a large amount of data (three groups and two modalities) there is little attempt to integrate these data together (e.g multivariate analysis / factor analysis) or even more simply to interpret the data together as a whole. This feels like a lost opportunity and the data is instead presented in long form and it is hard to pull out the most interesting findings from the noise. If this cannot be integrated, please make sure all data is deposited for others to analyse further. The discussion section appears overly speculative; a pathway description is not required for every metabolite seen in the study and most will not be necessary or mechanistic for this condition.

The remaining problems reading relate to ease of understanding the text and the presentation of the data. Suggestions are provided and require only minor revision.

Minor comments

You use the term “model” group, but this is ambiguous, I think actually mean control or sham or equivalent Please amend in all text.

Figure 1 – surgical images can be useful but the two right hand images are not necessary (microcurrent device and the low magnification rat and frame) – so please remove them. The two right hand images are useful but please add label with arrows and refer to each image in the figure legend. Note, in your methods you write that “neck skin depilated and disinfected with iodine tincture”, but this does not appear to have happen in the images? Please clarify.

Figure 3 – please add the group labels to the images directly for clarity and readability

Results 2.1 – Please explain the purpose of the experiment before you describe the data at the beginning of the paragraph, so the reader has context for what is to follow. Two sentences are sufficient.

Results 2.2 – Again, please explain the context of the experiment at the beginning of the paragraph before describing the data. This is otherwise quite hard to follow. Two sentences are sufficient.

Figure 4 – please add group labels to the images for clarity.

Suggestions – I think in Figures 3 and 4 it is awkward to compare down the columns. Would it not be easier to compare groups from left to right instead? You could rotate/ transform the images into this format. I leave this to author discretion.

Results 2.4.1 – again please explain the context of the experiment as in 2.1 and 2.2 above.

P22 - Some phrases are confusing in naming. E.g “In this study, SHR rats aged 13 to 14 weeks were selected, and the samples were collected 1 week after modelling” – is this the microcurrent damage? It is confusing that you have the other group called “model”. And on page 27 you refer to another group of “SHR model”. Please clarify in the text in each instance.

Discussion Figure 3 – this figure does not seem useful and is also not quantified. If the data is already in the results section, then it should not be duplicated. The name as Figure 3 also does not make sense as there are 7 figures above. Please either remove if it is essential then move it to the results.

Discussion text. This is overly speculative with long description of pathways with only passing relevance to the data in the manuscript e.g p26 GABA. Please cut down the discussion to include only those hypotheses most strongly supported by the data in this manuscript and the wider literature.

In a previous comment the authors say “All relevant data are within the manuscript and its Supporting Information files” however this does not appear to be in the submission. Please provide all raw data for this manuscript

**Reviewer #5: ** The authors responded to the the comments with care and diligence, making substantial revisions that improved the clarity, rigor, and overall quality of the manuscript. Methodological details were clarified, additional analyses were added, and key sections were rewritten for precision and coherence. The manuscript is now suitable for publication and makes a valuable contribution to the field.

**Do you want your identity to be public for this peer review?** For information about this choice, including consent withdrawal, please see our Privacy Policy

Reviewer #3: **Yes: ** keiichi abe

Reviewer #4: No

Reviewer #5: No

---

## [Author Response · Author response to Decision Letter 2]

28 Aug 2025

Reviewer #3:

while it is acceptable that hippocampal pathology was examined in this study, it would be helpful for the authors to clarify that:

Only the hippocampus was assessed histologically,

The behavioral evaluation mainly reflects cognitive performance and does not encompass depressive or affective domains,

The findings should be interpreted within these limitations.

Moreover, while the metabolomic and proteomic analyses are well conducted, many of the identified molecular changes—particularly those involving neurotransmitters and amino acid metabolism—are broadly linked to aging or psychiatric disorders. Some brief discussion on this point would strengthen the clinical relevance of the findings.

In summary, I recommend only minor textual revisions, particularly to the limitations and discussion sections, with the goal of making the implications more clinically accessible for practicing physicians. Overall, the manuscript is of high quality and should be considered favorably for publication after these small adjustments.

Sincerely thank you for your highly constructive and professional comments on this study. You have accurately identified the key areas that need improvement in the study’s evaluation scope and result interpretation, and these comments hold important guiding significance for enhancing the rigor, clinical relevance, and readability of the study. We have carefully organized your suggestions, supplemented the discussion on the study’s limitations, and streamlined the discussion section.

Reviewer #4: 

You use the term “model” group, but this is ambiguous, I think actually mean control or sham or equivalent Please amend in all text.

Thank you very much for your comments. The focus of this study is on proposing a new method for establishing a rat model of vascular cognitive impairment (VCI), and it is considered that "SHR + microcurrent stimulation" is a promising option for VCI model establishment. Therefore, SHR rats + microcurrent stimulation were defined as the Model group, for comparison with the other two groups. The WKY group consists of WKY rats that underwent only common carotid artery dissection without microcurrent stimulation; The SHR group consists of SHR rats without any intervention.

Figure 1 – surgical images can be useful but the two right hand images are not necessary (microcurrent device and the low magnification rat and frame) – so please remove them. The two right hand images are useful but please add label with arrows and refer to each image in the figure legend. Note, in your methods you write that “neck skin depilated and disinfected with iodine tincture”, but this does not appear to have happen in the images? Please clarify.

Thank you for your comment. The surgical images have been removed.

Results 2.1 – Please explain the purpose of the experiment before you describe the data at the beginning of the paragraph, so the reader has context for what is to follow. Two sentences are sufficient.Results 2.2 – Again, please explain the context of the experiment at the beginning of the paragraph before describing the data. This is otherwise quite hard to follow. Two sentences are sufficient.Results 2.4.1 – again please explain the context of the experiment as in 2.1 and 2.2 above.

Discussions on experimental objectives have been added to Sections 2.1, 2.2 and 2.4.

Figure 3 – please add the group labels to the images directly for clarity and readability Figure 4 – please add group labels to the images for clarity.

Thank you very much for your comment. Group labels have been directly added to the pathological images.

P22 - Some phrases are confusing in naming. E.g “In this study, SHR rats aged 13 to 14 weeks were selected, and the samples were collected 1 week after modelling” – is this the microcurrent damage? It is confusing that you have the other group called “model”. And on page 27 you refer to another group of “SHR model”. Please clarify in the text in each instance.

Thank you for your valuable comments. Here, the SHR rats specifically refer to those in the SHR group. We apologize for any confusion caused. Additionally, all mentions of the "SHR group" and "Model group" in the manuscript have been revised to "SHR group (no vascular injury)" and "Model group (SHR+vascular injury composite model)" respectively.

Discussion Figure 3 – this figure does not seem useful and is also not quantified. If the data is already in the results section, then it should not be duplicated. The name as Figure 3 also does not make sense as there are 7 figures above. Please either remove if it is essential then move it to the results.

Thank you for your suggestion. The table has been moved to the Results section.

Discussion text. This is overly speculative with long description of pathways with only passing relevance to the data in the manuscript e.g p26 GABA. Please cut down the discussion to include only those hypotheses most strongly supported by the data in this manuscript and the wider literature.

Sincerely thank you for your incisive comments and constructive suggestions on the Discussion section of this study. Regarding this issue, the Discussion section has undergone systematic streamlining and restructuring: specifically, we have eliminated subjective speculations, condensed redundant pathway descriptions, and retained only the hypotheses supported by the data of this study.

In a previous comment the authors say “All relevant data are within the manuscript and its Supporting Information files” however this does not appear to be in the submission. Please provide all raw data for this manuscript

Thank you for your reminder. The data have been supplemented.

Reviewer #5: 

The authors responded to the the comments with care and diligence, making substantial revisions that improved the clarity, rigor, and overall quality of the manuscript. Methodological details were clarified, additional analyses were added, and key sections were rewritten for precision and coherence. The manuscript is now suitable for publication and makes a valuable contribution to the field.

Sincerely thank you for your recognition and affirmation of the revision work of this study! It is precisely based on your previous professional comments that we have been able to targetedly clarify the methodological details, ultimately enabling the paper to better align with academic standards and better highlight its research value. Once again, we extend our most sincere gratitude to you! Your professional guidance has been key support for improving the quality of this study, and has also set a benchmark for rigorous academic research in our future scientific work.

---

## [Decision Letter · Decision Letter 2]

5 Sep 2025

Combined metabolomics and proteomics analysis of vascular cognitive impairment in hypertensive rats induced by endothelial injury

PONE-D-24-59603R2

Dear Dr. Liu,

We’re pleased to inform you that your manuscript has been judged scientifically suitable for publication and will be formally accepted for publication once it meets all outstanding technical requirements.

Kind regards,

Nafisa M. Jadavji, PhD, MSc, BSc

Academic Editor

PLOS ONE

Additional Editor Comments (optional):

Reviewer #3:

Reviewer #5:

Reviewers' comments:

Reviewer's Responses to Questions

**Comments to the Author**

Reviewer #3: All comments have been addressed

Reviewer #5: All comments have been addressed

2. Is the manuscript technically sound, and do the data support the conclusions?

Reviewer #3: Yes

Reviewer #5: Yes

3. Has the statistical analysis been performed appropriately and rigorously?

Reviewer #3: Yes

Reviewer #5: Yes

4. Have the authors made all data underlying the findings in their manuscript fully available?

Reviewer #3: Yes

Reviewer #5: Yes

5. Is the manuscript presented in an intelligible fashion and written in standard English?

Reviewer #3: Yes

Reviewer #5: Yes

Reviewer #3: Since the authors have explicitly stated that this work is a preliminary study, it seems more appropriate not to extend the discussion too broadly to the relevance of the proteome across multiple other diseases.

Taking this into consideration, I find that the authors have adequately addressed the reviewers’ comments overall, and the manuscript has been sufficiently improved.

Reviewer #5: As I commented in previous versions of the article, the paper can be accepted for publication in PLOS ONE

**Do you want your identity to be public for this peer review?** For information about this choice, including consent withdrawal, please see our Privacy Policy

Reviewer #3: **Yes: ** Keiichi Abe

Reviewer #5: No

---

## [Editor Report · Acceptance letter]

PONE-D-24-59603R2

PLOS ONE

Dear Dr. Liu,

I'm pleased to inform you that your manuscript has been deemed suitable for publication in PLOS ONE. Congratulations! Your manuscript is now being handed over to our production team.

Kind regards,

on behalf of

Dr. Nafisa M. Jadavji

Academic Editor

PLOS ONE